# Molecular basis for the inhibition of the methyl-lysine binding function of 53BP1 by TIRR

Jiaxu Wang[1], Zenglin Yuan [2], Yaqi Cui[1,3], Rong Xie[1,3], Guang Yang[3], Muzaffer A. Kassab[3], Mengxi Wang[1], Yinliang Ma[1,3], Chen Wu[1], Xiaochun Yu[3] & Xiuhua Liu[1]

53BP1 performs essential functions in DNA double-strand break (DSB) repair and it was recently reported that Tudor interacting repair regulator (TIRR) negatively regulates 53BP1 during DSB repair. Here, we present the crystal structure of the 53BP1 tandem Tudor domain (TTD) in complex with TIRR. Our results show that three loops from TIRR interact with 53BP1 TTD and mask the methylated lysine-binding pocket in TTD. Thus, TIRR competes with histone H4K20 methylation for 53BP1 binding. We map key interaction residues in 53BP1 TTD and TIRR, whose mutation abolishes complex formation. Moreover, TIRR suppresses the relocation of 53BP1 to DNA lesions and 53BP1-dependent DNA damage repair. Finally, despite the high-sequence homology between TIRR and NUDT16, NUDT16 does not directly interact with 53BP1 due to the absence of key residues required for binding. Taken together, our study provides insights into the molecular mechanism underlying TIRR-mediated suppression of 53BP1-dependent DNA damage repair.

[1] College of Life Sciences, Hebei University, Baoding 071000 Hebei, China. [2] State Key Laboratory of Microbial Technology, Shandong University, Jinan 250100 Shandong, China. [3] Department of Cancer Genetics and Epigenetics, Beckman Research Institute, City of Hope, 1500 East Duarte Road, Duarte, CA 91010, USA. These authors contributed equally: Jiaxu Wang, Zenglin Yuan, Yaqi Cui. Correspondence and requests for materials should be addressed to X.Y.(email: xyu@coh.org) or to X.L.(email: liuxiuhua_2004@163.com)

Cells constantly encounter genotoxic stress that induces DNA double-strand breaks (DSBs). To repair DSBs, cells have evolved a sophisticated DSB repair system and p53-binding protein 1 (53BP1) plays a predominant role in DSB repair; thereby promoting genomic stability[1,2].

In response to DSBs; PI3 like kinases, including ATM, ATR, and DNAPK, phosphorylate H2AX in the vicinity of the DSB and initiate a signaling cascade, which leads to RNF8 and RNF168-mediated ubiquitylation of chromatin[3]. These molecular events induce the recruitment of 53BP1 to DNA lesions. The minimal focus-forming region (FFR) of 53BP1 required for localization of 53BP1 to DSBs consists of an oligomerization domain, a tandem Tudor domain (TTD) and the ubiquitin-dependent recruitment (UDR) motif[4,5]. The TTD binds dimethylated lysine 20 on histone H4 (H4K20me2) while UDR binds the mono-ubiquitylated lysine 15 on histone H2A[6]. Besides H4K20me2, 53BP1 TTD may also recognize dimethylated lysine 810 on tumor suppressor protein pRb and dimethylated lysine 370 and 382 on the tumor suppressor p53[7–9].

It has been reported that the DNA damage repair functions of 53BP1 are dependent on its recruitment to DSBs via recognition of H4K20me2, which is also the most abundant histone lysine methyl mark, present in around 85% of all histone H4 molecules[8,10]. Thus, it is necessary to suppress the binding of 53BP1 on the chromatin during normal cellular functions and counter these regulations by unmasking H4K20me2, thereby enabling 53BP1 binding when DSBs occur. This is accomplished by the binding of tandem Tudor motifs on JMJD2A/B to H4K20me2, and DNA damage triggers degradation of JMJD2A/B thereby allowing the exposure of methylated H4K20 for the binding of 53BP1[11]. In addition, an alternative 53BP1 recruitment pathway involves the release of a polycomb protein L3MBTL1 from H4K20me2 due to the ATPase activity of valosin-containing protein (VCP) following DNA damage[12]. Thus, both pathways involve proteins that particularly bind to H4K20me2 before 53BP1 recruitment and represent indirect mechanisms, i.e., restricting 53BP1 access to chromatin. Notably, recent studies propose a direct regulatory mechanism in which Tudor interacting repair regulator (TIRR, aka NUDT16L1) specifically binds to the 53BP1 TTD, forming a stable TIRR–53BP1 complex and consequently regulating the recruitment of 53BP1 on chromatin[13,14].

TIRR belongs to the NUDIX hydrolase family, and shares 46% sequence identity with NUDT16. It has been shown that NUDT16 is a pyrophosphatase to decap mRNA or hydrolyze other nucleic acid substrates[15–19]. However, TIRR lacks key enzymatic residues required for the hydrolysis of phosphodiester bond and consequently cannot participate in hydrolysis. Although, TIRR lacks enzymatic activity, it plays a significant role in regulating 53BP1 pathway and function via its interaction with 53BP1 TTD[13,14]. Thus, depletion or overexpression of TIRR impairs 53BP1-dependent function in DSB repair[13,14]. Moreover, TIRR amplification in human cancer cell lines[13] abolishes the recruitment of 53BP1 and its downstream effector RIF1 to DSB sites, thus disrupting 53BP1-dependent DSB repair[13]. In contrast, depletion of TIRR destabilizes the nuclear-soluble fraction of 53BP1[14]. Thus, during DSB, TIRR exerts a two pronged regulatory effect to suppress the function of 53BP1. However, in undamaged cells, TIRR directly binds 53BP1, preventing its interaction with H4K20me2. This interaction preserves 53BP1 stabilization and its sub-nuclear localization[13]. Once genomic DNA is damaged, 53BP1 is released from the 53BP1–TIRR complex, which allows 53BP1 to recognize H4K20me2 and enter the following DNA repair pathway[13].

Although the role of TIRR in modulating 53BP1 function has been reported[13,14], the molecular mechanism by which TIRR interacts with the TTD of 53BP1 remains unknown. Here, we present 2.0 Å resolution crystal structure of the human 53BP1 TTD in complex with TIRR, and elucidate the structural basis how TIRR suppresses the interaction between 53BP1 TTD and H4K20me2.

## Results

**The crystal structure of TIRR and 53BP1 TTD.** In order to understand the molecular mechanism by which TIRR modulates 53BP1 function in response to DNA damage, we determined the crystal structure of TIRR and 53BP1 TTD complex at 2.0 Å resolution by X-ray diffraction (Fig. 1a). TIRR contains six α-helices and six β-strands, and resembles the canonical NUDIX fold, which is composed of a typical α/β/α sandwich (Fig. 1b)[20]. The TTD in the TIRR–53BP1 TTD complex adopts a typical TTD fold conformation, containing two β-barrels and one C-terminal α-helix. This conformation closely resembles the apo-form of the TTD (PDB ID: 2G3R). The r.m.s. deviation (RMSD) between apo-TTD and the TTD in the TIRR–53BP1 complex is 0.648 Å for all corresponding Cα atoms. Although the two structures are very similar, subtle difference was found in the loop ($Lys^{1494}$ - $Phe^{1501}$) between β1 and β2 (Fig. 1c). A closer inspection of these two structures reveals that the $Trp^{1495}$ and $Tyr^{1523}$ from the dimethyl-lysine binding pocket (H4K20me2 binding cage) take different conformation (Fig. 1d).

**The binding interface between 53BP1 TTD and TIRR.** The TTD of 53BP1 is known to recognize dimethyl-lysine[8]. The dimethyl-lysine binding pocket of TTD is composed of $Asp^{1521}$ and four aromatic residues ($Trp^{1495}$, $Tyr^{1502}$, $Phe^{1519}$, $Tyr^{1523}$) all contributed from the β-barrel 1. In the crystal structure of the TIRR–53BP1 TTD complex, TIRR interacts with the TTD around the dimethyl-lysine binding pocket (Fig. 2a). TIRR and 53BP1 TTD share a total buried surface area of 676.8 $Å^2$. The binding interface of TIRR with the TTD primarily consists of the N-terminus loop (residues 6–14), the loop between α1 and β1 (residues 19–25-namely α1-β1 loop) and the loop between β4 and β5 (residues 101–108-namely β4-β5 loop). Extensive hydrogen bonds and hydrophobic interactions are formed at the interface, mediating the stabilization of binding interaction between TIRR and TTD.

Residue $Lys^{10}$ from the N-terminus loop of TIRR inserts into the TTD, and is stabilized by water-mediated hydrogen bond network formed between $Gln^{11}$ and $Lys^{10}$ residues of TIRR, a $H_2O$ molecule, and residues $Tyr^{1523}$ and $Trp^{1495}$ of 53BP1 TTD (Fig. 2b). In contrast to TIRR in the complex with 53BP1 TTD, the apo-TIRR lacks the presence of clear electron density in the side chain of $Lys^{10}$ (PDB ID: 3KVH) (Supplementary Fig. 1a). It indicates that $Lys^{10}$ of TIRR is relatively more flexible and in disorder. However, $Lys^{10}$ of TIRR becomes more stable upon interaction with 53BP1.

Three residues ($Pro^{105}$-$His^{106}$-$Arg^{107}$) of the β4-β5 loop of TIRR participate in the binding to 53BP1 TTD via hydrophobic interactions and hydrogen bonds (Fig. 2c). The hydrophobic residue $Pro^{105}$ was clamped by hydrophobic residues $Tyr^{1500}$ and $Phe^{1553}$ of 53BP1 TTD. The $His^{106}$-Nε2 forms hydrogen bonds with the hydroxyl groups of $Glu^{1551}$ and $Tyr^{1552}$ of 53BP1 TTD. $Arg^{107}$ of TIRR forms hydrogen bonds with $Met^{1584}$ and $Asp^{1521}$ of the TTD. Moreover, crystal structure of the apo-TIRR (PDB ID: 3KVH), lacks clear electron density at four residues ($Glu^{103}$-$Gly^{104}$-$Pro^{105}$-$His^{106}$) in the β4-β5 loop, suggesting that this loop is in disorder conformation (Supplementary Fig. 1b, c). Taken together, it indicates that this loop becomes more oriented once TIRR binds to 53BP1.

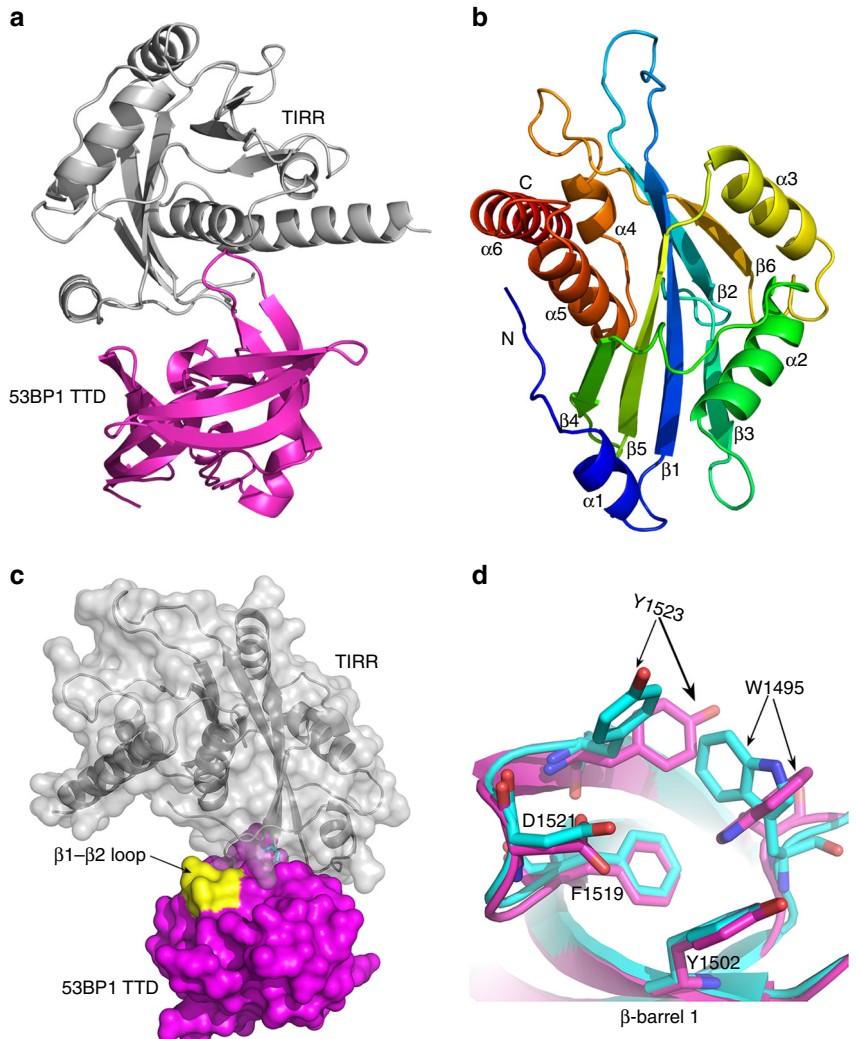

**Fig. 1** The co-structure of the TIRR–53BP1 TTD complex. **a** Cartoon representation of the crystal structure of the TIRR–53BP1 TTD complex. **b** Rainbow colored cartoon representation of TIRR. Six α-helices and six β-strands are labeled. **c** Molecular surface representation of the TIRR-53BP1 TTD heterodimer. TIRR is gray, 53BP1 TTD is magenta, and the subtle difference in the loop between β1 and β2 in 53BP1 TTD is displayed in yellow color. **d** Superposition of β-barrel 1 from 53BP1 TTD in the structure of the TIRR–53BP1 TTD complex with that of the H4K20me2–53BP1 TTD complex (PDB ID: 2IG0). The TTD in the structure of the TIRR–53BP1 TTD complex is rendered as magenta cartoon, and the TTD in the H4K20me2–53BP1 TTD complex is rendered as cyan cartoon. The amino acids corresponding to the dimethyl-lysine binding pocket are represented as stick

Different from the N-terminus loop and the β4-β5 loop, the α1-β1 loop does not form any hydrogen bond with 53BP1 TTD. Instead, two residues (Leu[20] and Trp[24]) from the α1-β1 loop, Ile[12] from the N-terminus loop along with Leu[101] and Val[109] from the β4-β5 loop form a hydrophobic layer covering the hydrophobic dimethyl-lysine binding pocket. These interactions generate a hydrophobic core at the binding interface to stabilize the TIRR–TTD complex (Fig. 2d). As mentioned above, major structural variation in the TTD within the H4K20me2–53BP1 and TIRR–53BP1 complex is present in the loop (residues 1494–1501) between β1 and β2. In case of interaction with H4K20me2, three residues Ser[1496]-Ser[1497]-Asn[1498] in the TTD adopt an inward-facing conformation at the dimethyl-lysine binding pocket, which facilitates binding of H4K20me2[8]. Notably, these same residues in the TTD within the TIRR–53BP1 complex are in outward-facing conformation (Fig. 2e). It is likely that residues of TIRR and TTD surrounding this binding interface are hydrophilic, and a hydration shell is inserted into this interface leading to the conformational change. Nevertheless, TIRR is able to abolish the interaction between 53BP1 and

H4K20me2 due to its masking of the dimethyl-lysine binding pocket of 53BP1 TTD (Fig. 2d).

**In-vitro binding analysis of the TIRR–53BP1 TTD complex**. To validate the interaction between TIRR and 53BP1 TTD, we performed ITC assays and found that the dissociation constant between these proteins (Kd) is ~ 0.78 μM (Fig. 3a). The binding affinity is remarkably higher compared to H4K20me2 peptide and 53BP1 TTD (Fig. 3b). When competition pull-down assays were performed using equimolar peptides, TIRR was able to largely compete out H4K20me2 peptide and bind to 53BP1 TTD (Fig. 3c), demonstrating its ability to suppress 53BP1 in cells.

Multiple sequence alignment shows that the crucial residues of TIRR involved in the binding with 53BP1 including Lys[10], Pro[105], His[106], and Arg[107] are highly conserved in vertebrates (Supplementary Fig. 2). To demonstrate the significance of these residues in mediating the interaction, recombinant TIRR mutants K10E, P105A, H106A, and R107A were generated. These mutations abolished the binding interaction between TIRR and 53BP1 TTD

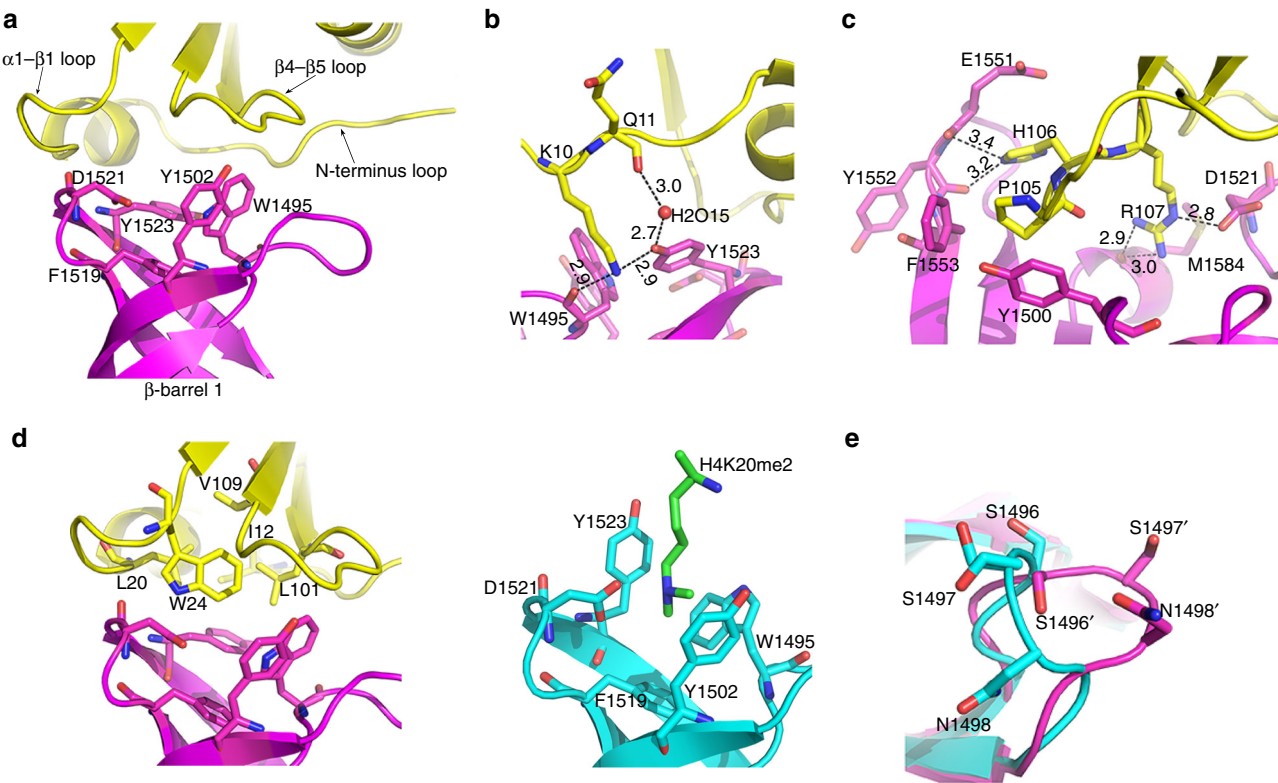

**Fig. 2** Detailed view of TIRR (yellow) binding to 53BP1 TTD (magenta). **a** Three loops from TIRR are involved in the interaction with the dimethyl-lysine binding pocket of 53BP1 TTD. The three loops are labeled as N-terminus loop, α1-β1 loop, and β4-β5 loop respectively. The dimethyl-lysine binding pocket is from the β-barrel 1 of the TTD, the five residues representing the pocket are labeled. **b** Interaction details of TIRR Lys[10] with 53BP1 TTD. Lys[10] and Gln[11] from TIRR are in yellow stick, Tyr[1523] and Trp[1495] from the TTD are in magenta stick, and the water molecule is in red sphere. The hydrogen bonds are shown as dashed lines. **c** Interaction details of TIRR Pro[105]-His[106]-Arg[107] with 53BP1 TTD. The three residues Pro[105], His[106], and Arg[107] from TIRR are displayed in yellow stick. The corresponding interacting residues from TTD are shown in magenta stick. The hydrogen bonds are shown as dashed lines. **d** Structural comparison of TIRR and H4K20me2 binding to the β-barrel 1 of 53BP1 TTD. Left: The TIRR hydrophobic layer consisting of five residues covers the hydrophobic dimethyl-lysine binding pocket of 53BP1 TTD. Right: H4-K20me2 (green stick) binds to the dimethyl-lysine binding pocket of 53BP1 TTD (PDB ID: 2IG0). The residue H4-K20me2 is in green stick, and 53BP1 TTD is in cyan cartoon. **e** The subtle differences in the loop between β1 and β2 from the TTD molecule in the TIRR–53BP1 TTD complex compared to that in the H4K20me2–53BP1 TTD complex (PDB entry 2IG0, cyan). Ser[1496], Ser[1497], and Asn[1498] are shown in stick

in the in-vitro protein pull-down assays (Fig. 3d). Moreover, based on the structural analysis, we also mutated key residues in the TTD including W1495A, Y1500A, D1521A, and Y1523A. Again these mutations abolished the binding between TIRR and 53BP1 TTD (Fig. 3e). We also confirmed the binding affinities of these mutants using ITC assays, which were consistent with the results of the protein pull-down assays (Supplementary Fig. 3).

TIRR is also known as NUDT16L1 and shares 46% homology with NUDT16, a member of the NUDIX enzyme super family that hydrolyzes phosphodiester bond for mRNA decapping and nucleic acid metabolism[17–19]. Structural superposition between TIRR and NUDT16 gave RMSD value of 1.36 Å for all equivalent Cα atoms, indicating overall similarity in their conformations. However, there are significant differences in the key residues (Lys[10] and His[106]) between TIRR and NUDT16. Lys[10] in TIRR is replaced by an Arg residue in NUDT16, His[106] in TIRR is missing in NUDT16 (Supplementary Figs. 2 and 4). Consistent with these observations, there was no direct interaction between NUDT16 and 53BP1 TTD as demonstrated by in-vitro pull-down assays (Supplementary Fig. 4).

**TIRR regulates 53BP1-dependent DSB response.** Next, based on in-vitro structural analysis, we examined the functional significance of the interaction between TIRR and 53BP1. Based on the observation that TIRR is overexpressed in cancer cells[13],

wild-type TIRR and TIRR mutants (K10E, P105A, H106A, and R107A) were expressed in 293T cells. In agreement with the in-vitro structure analysis and pull-down assays, only wild-type TIRR but not the mutants was able to co-immunoprecipitate (co-IP) with 53BP1 (Fig. 4a).

It has been shown that TIRR acts as a negative regulator of DSB repair pathway by sequestering 53BP1 in the nucleoplasm and suppressing its relocation to DSBs[13,14]. Consistent with co-IP results, we observed that key residues of TIRR (Lys[10], Pro[105], His[106], and Arg[107]) involved in interaction with 53BP1 were required for the inhibitory function of TIRR. Ectopic over-expression of K10E, P105A, H106A, or R107A could not abolish the formation of ionizing radiation-induced foci (IRIF) of 53BP1, whereas wild type was able to do so (Fig. 4b). Moreover, 53BP1 is a key regulator in the DSB response pathway and mediates the recruitment of RIF1 to the sites of DSBs. Thus, we further explored the IRIF of RIF1 and found that only wild-type TIRR, but not the TIRR mutants lacking key interacting residues, can abolish the IRIF of RIF1 (Fig. 4c). Collectively, these results suggest that interaction between TIRR and 53BP1 is important for regulating 53BP1-dependent DSB response.

**Dimer formation of TIRR.** Notably, the crystal structure of TIRR–53BP1 complex consists of one TTD and two TIRR

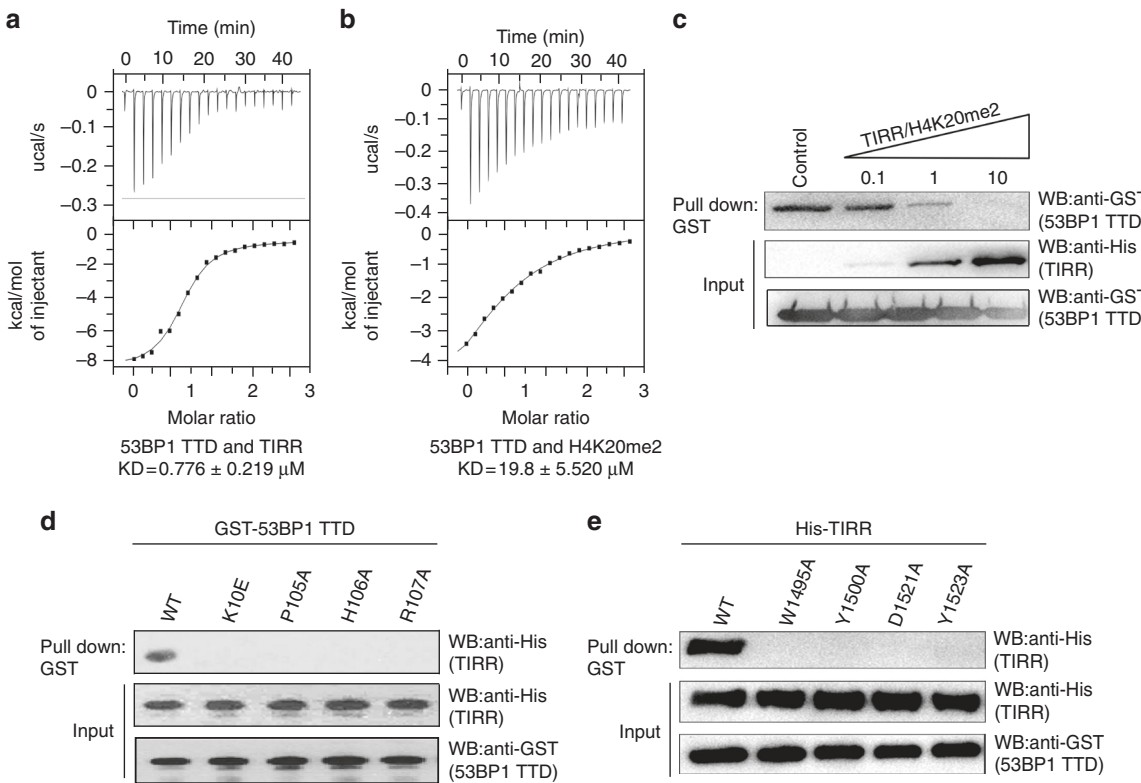

**Fig. 3** The in-vitro binding analysis of the TIRR–53BP1 TTD complex. **a** The ITC titration of wild-type TIRR with 53BP1 TTD. **b** The ITC titration of 53BP1 TTD with H4K20me2 peptide (residues 12–25). **c** TIRR competes out H4K20me2 peptide (residues 12–25) for the interaction with 53BP1 TTD. 0.1, 1, and 10 represent the molar ratio between TIRR and H4K20me2 in each sample. **d** The mutations of TIRR abolish the interaction with 53BP1 TTD. The in-vitro interaction between TIRR and 53BP1 TTD was examined by the GST pull-down assay and western blot with indicated antibodies. **e** The mutations of 53BP1 TTD abolish the interaction with TIRR. Uncropped blots are shown in Supplementary Fig. 10

molecules (Supplementary Fig. 5a). PISA analysis reveals that the total buried surface between two TIRR molecules is ~ 3230 Å$^2$, which is about 17.3% of the accessible surface area of these molecules. The RMSD between two monomers from this complex is 0.638 Å for all equivalent Cα atoms, indicating that there are no obvious conformational differences between the two TIRR monomers (Supplementary Fig. 5b).

Moreover, using ITC measurements, we found that the binding stoichiometry of the interaction between TIRR and 53BP1 TTD is 1:1. We also performed analytical ultracentrifugation to investigate the existence of the TIRR–TTD complex in solution. We observed that TIRR existed as a stable dimer (MW: 44.5 kDa), while TTD existed as a monomer (MW: 19.3 kDa) in solution. Although TTD existed as a monomer, the calculated MW of the TIRR–TTD complex in solution was 81.5 kDa, corresponding to presence of 2TIRR: 2TTD (Supplementary Fig. 6). Thus, analytical ultracentrifugation analysis indicates that TIRR homodimer binds two TTD monomers in solution. However, our crystal structure analysis of TIRR–TTD complex suggests that only one TTD molecule binds to TIRR homodimer, and TIRR conformation is only slightly affected due to this binding. This phenomenon could result from space hindering in crystal packing. This assumption was supported by artificial modeling-artificial binding of one more TTD molecule to another TIRR monomer led to the collapse of the modeled TTD molecules due to serious steric hindrance in the crystal lattice (Supplementary Fig. 7a). Thus, in order to maintain thermal stability in the crystal packing, TIRR homodimer would reasonably bind one TTD, rather than two molecules. Nevertheless, the 53BP1 TTD–TIRR complex is likely to contain two 53BP1 TTD and two TIRR molecules (Supplementary Fig. 7b).

Besides TIRR, it has been indicated that NUDT16 may also associate with 53BP1[14]. However, due to lacking of the key binding residues, NUDT16 cannot directly interact with 53BP1. The NUDIX hydrolase family protein tends to form dimer in solution, given the sequence identity, similarity on overall conformation and dimer interface, it is possible that TIRR and NUDT16 form a heterodimer. In fact, TIRR was able to co-IP with NUDT16 and interact with NUDT16 in-vitro (Supplementary Figs. 8 and 9a). Based on the structural analyses, we generated different mutations in TIRR to abolish the TIRR homodimer or TIRR/NUDT16 heterodimer. With the pull-down screening, we found that triple-mutation (L60Y/V143Y/F160A) at the dimer interface abolished the dimer formation (Supplementary Fig. 9a, e, and f). Moreover, these mutations did not affect the interaction with 53BP1 because the dimer interface is far away from the interaction sites with 53BP1 (Supplementary Fig. 9b, e). However, disrupting the dimer formation destabilized 53BP1 in the cell, suggesting that the dimer formation plays a key role to maintain the complex stability[21] (Supplementary Fig. 9c).

**TIRR regulates DSB repair.** Accumulated evidence suggests that 53BP1 is involved in non-homologous end-joining (NHEJ)-an error prone type of DSB repair[1]. Notably, loss of 53BP1 promotes homologous recombination (HR), which is an error free type of DSB repair in the BRCA1-deficient cells[2,22]. As BRCA1 participates in HR repair, loss of BRCA1 abolishes HR[23]. In the presence of ectopically expressed TIRR, 53BP1 will be trapped away from DSB sites, thus promoting HR for DSB repair in the BRCA1-deficient cells[13]. Since the first step of HR is to process DSB into single stranded DNA overhang that is immediately

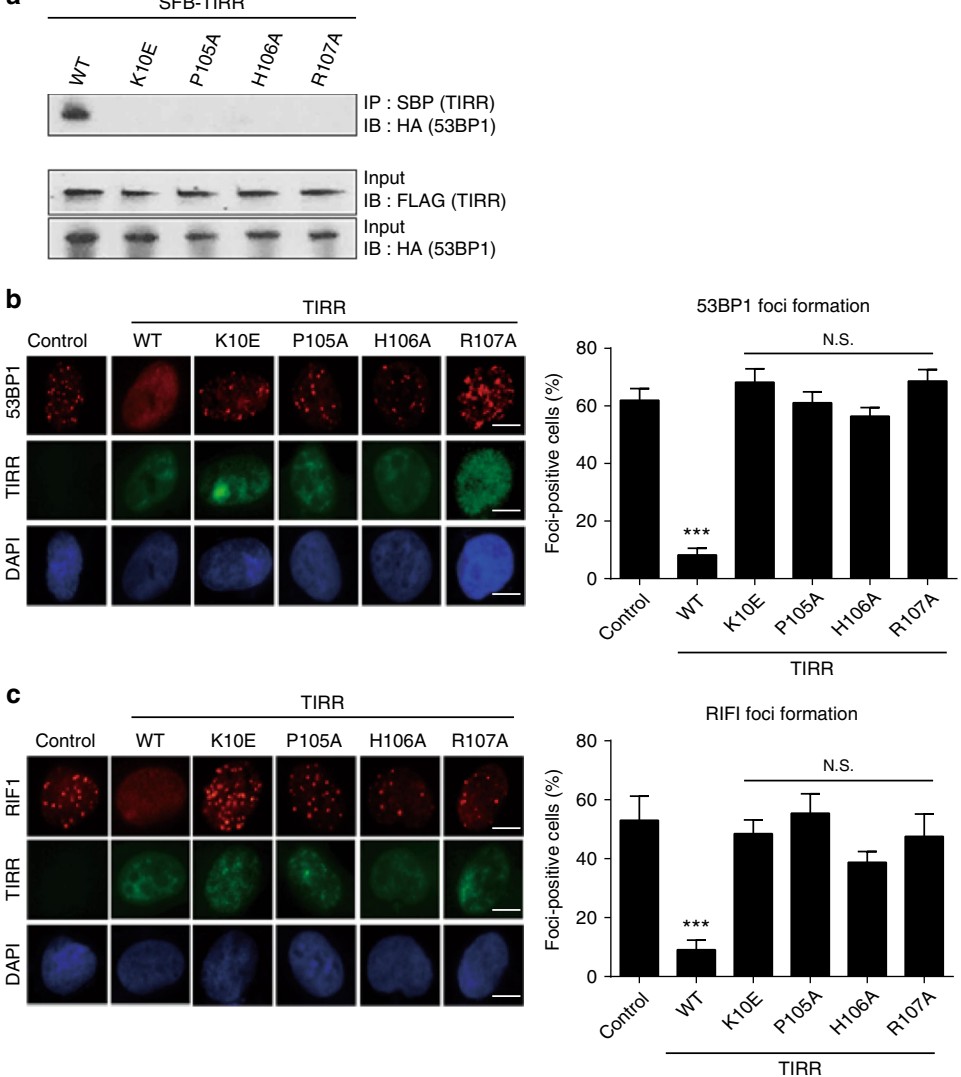

**Fig. 4** TIRR regulates 53BP1-dependent DSB response. **a** The mutations of TIRR disrupt the TIRR–53BP1 complex in vivo. TIRR, TIRR mutants, and 53BP1 were expressed in 293T cells. Co-IP and western blot were examined with indicated antibodies. **b** The mutations of TIRR do not affect the IRIF of 53BP1. Wild-type TIRR or its mutants were expressed in U2OS cells. The cells were treated with 10 Gy of IR. The IRIF of endogenous 53BP1 was examined by immunofluorescence staining. Foci positive cells were examined from three independent experiments with 100 randomly selected cells in each experiment. Scale bar represents 5 μm. **c** The mutations of TIRR do not affect the IRIF of RIF1. Data are represented as mean ± s.d. as indicated from three independent experiments. N.S.: non significant; ***: statistically significant ($p < 0.001$). Scale bar represents 5 μm

coated by the RPA complex, we examined the resection of DSBs using phospho-RPA as the surrogate maker in the BRCA1-deficient cells. We observed that wild-type TIRR promoted the IRIF of phospho-RPA; however, none of the mutants were able to facilitate the IRIF of phospho-RPA, indicating that the interaction between TIRR and 53BP1 may play a role to promote HR repair in the BRCA1-deficient cells (Fig. 5a). Moreover, we examined a downstream effector RAD51, a key recombinase in HR, and found that only wild-type TIRR but not the mutants facilitated the IRIF of RAD51 (Fig. 5b). This observation further confirms that the interaction between TIRR and 53BP1 is important for the HR activation in BRCA1-deficient cells.

In addition, BRCA1-deficient cells are hypersensitive to PARP inhibitor treatment due to defects in HR[24]. When TIRR was ectopically expressed in BRCA1-deficient cells, they regained HR and were less sensitive to PARP inhibitor treatment. However, ectopic overexpression of TIRR mutants did not affect the sensitivity of BRCA1-deficient cells to PARP

inhibitor treatment (Fig. 5c). Moreover, disrupting the dimer formation in BRCA1-deficient cells induced cells less sensitive to PARP inhibitor because it destabilized 53BP1 in the cells (Supplementary Fig. 9c, d). Taken together, our study demonstrates that TIRR acts as a negative regulator of 53BP1 during DSB repair.

## Discussion

Based on the analysis of NMR spectrum, a previous study has indicated that $Lys^{10}$ of TIRR is required for the binding to 53BP1[13]; however, the structure of the complex has not been solved. In this study, using X-ray diffraction, we have determined the structure basis of TIRR–53BP1 complex interaction. We found that multiple amino-acid residues from the three loops of TIRR contact with the TTD of 53BP1. In particular, these residues from TIRR cover the methyl-lysine binding pocket of the TTD, thus abolishing the interaction between 53BP1 and H4K20me2. These observations suggest that TIRR acts as a negative regulator of 53BP1. In

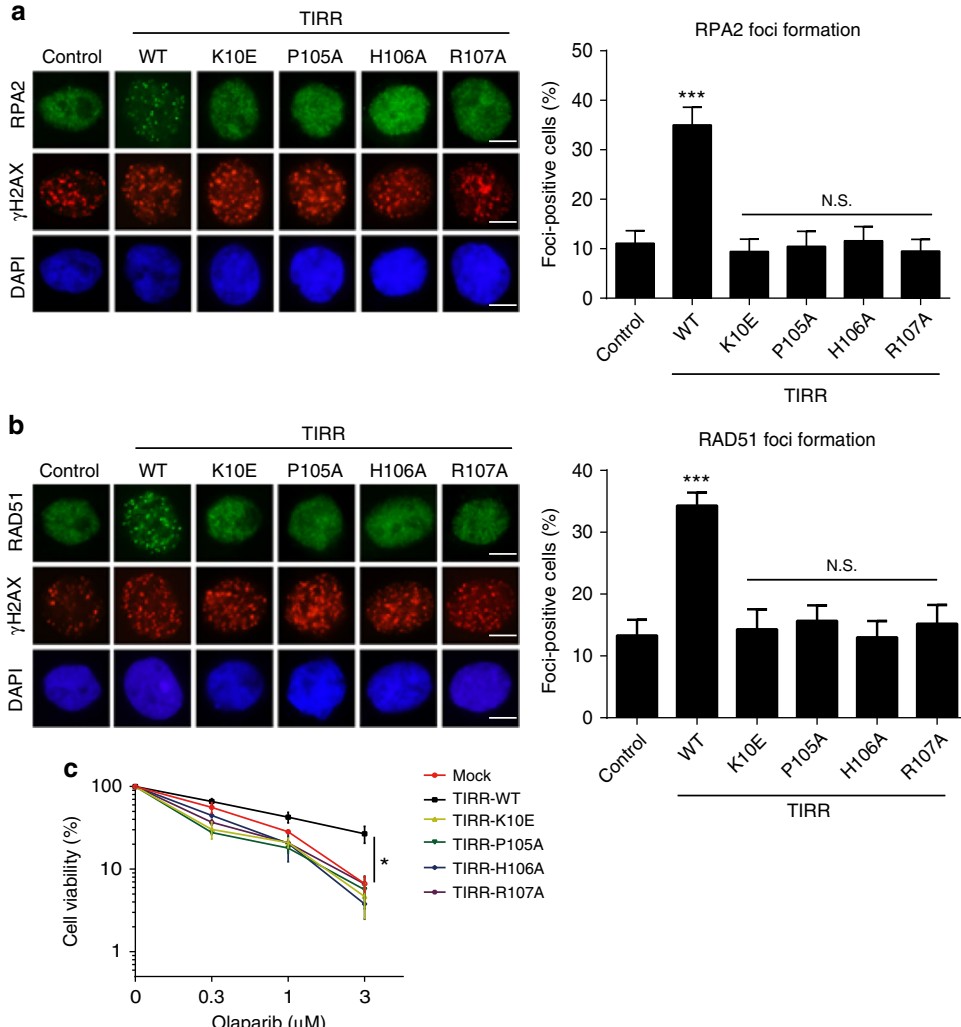

**Fig. 5** TIRR regulates DSB repair in the BRCA1-deficient cells. Wild-type TIRR or its mutants were expressed in UWB1 (BRCA1-deficient) cells. The cells were treated with 10 Gy of IR. The IRIF of RPA (**a**) or RAD51 (**b**) was examined by immunofluorescence staining. Alternatively, the cells were treated with indicated dose of olaparib (PARP inhibitor) for 6 days. Scale bar represents 5 μm. Cell viability was examined (**c**). Data are represented as mean ± s.d. as indicated from three independent experiments. N.S.: non significant; *: statistically significant ($p < 0.05$); ***: statistically significant ($p < 0.001$). Scale bar represents 5 μm

addition, we have comprehensively analyzed and mapped the key residues involved in TIRR–TTD interaction, and demonstrated that mutations of these key residues on either side would abolish the complex formation. Thus, our result has revealed the molecular basis for the TIRR–53BP1 complex formation.

A recent study also shows that *TIRR* locus is often amplified in human cancers[13]. Overexpression of TIRR abolishes the interaction of 53BP1 and H4K20me2, consequently suppressing the relocation of 53BP1 to DNA lesions, thus disrupts 53BP1-dependent DNA damage repair[13]. This unique molecular signature may impact cancer treatment with PARP inhibitors, given their expanding clinically use against BRCA-deficient tumors. Considering that loss of 53BP1 promotes HR repair and suppresses cellular sensitivity to PARP inhibitors, overexpression of TIRR in cancer cells may severely compromise 53BP1 function and increase the resistance of BRCA-deficient cancer cells to PARP inhibitors. Thus, TIRR can act as a novel biomarker to predict the efficacy of PARP inhibitors and consequently evolve the patient selection criteria in the clinical treatment of BRCA-deficient tumor patients. Moreover, as 53BP1 is able to regulate the tumor cell sensitivity to PARP inhibitor in BRCA tumors, our structure analysis on the complex of TIRR and 53BP1 provides the opportunities for developing chemical probes to

regulate the activity of 53BP1, which may facilitate clinical cancer treatment in future.

## Methods

**Protein expression and purification**. The DNA sequences encoding human TIRR (residues 6–211) and 53BP1 tandem Tudor domain (residues 1459–1634) were amplified from a 293T cDNA library and were subcloned into the modified pET-15b vector, which carries an N-terminal hexahistidine tag and a human rhinovirus 3C protease-cleavage site. The PCR primers used in this study were provided in Supplementary Tables 1 and 2.

*E. coli* BL21 (DE3) cells (in house) containing the plasmid pET-15b-TIRR were grown in LB broth media at 37 °C to an $OD_{600}$ of 0.8. Protein expression was induced by adding 0.12 mM IPTG at 22 °C for 16 h. The cells were harvested by centrifugation at $14,000 \times g$ for 15 min. The cell were lysed by sonication after resuspending in ice-cold lysis buffer (25 mM Tris-HCl, pH 8.0, 400 mM NaCl and 5% glycerol). The cell lysate was then centrifuged for 50 min at $30,000 \times g$. The supernatant containing recombinant protein TIRR was loaded onto a Ni-chelating Sepharose (GE Healthcare) column pre-equilibrated with lysis buffer. After extensive washing with lysis buffer, all bound protein was eluted using elution buffer (25 mM Tris-HCl, pH 8.0, 400 mM NaCl, 5% glycerol, and 250 mM imidazole). The N-terminal hexahistidine tag was removed by the 3C protease. The elutes were further purified using a Superdex 200 Increase column (GE Healthcare) in 10 mM Tris-HCl, pH 8.0, 200 mM NaCl and 3 mM DTT.

To purify the recombinant 53BP1 TTD protein, BL21 (DE3) cells containing plasmids encoding 53BP1 TTD was incubated in LB media. The protein induction, expression, and lysis conditions were same to that of BL21 (DE3) cells harboring

pET-15b-TIRR, except that 25 mM Tris-HCl, pH 8.0, and 100 mM NaCl was used as the lysis buffer. 53BP1 TTD was first purified by Ni-chelating Sepharose column using elution buffer (25 mM Tris-HCl, pH 8.0, 100 mM NaCl, 250 mM imidazole), and was then incubated with 3C protease to remove the N-terminal hexahistidine tag. The protein TTD was further purified by ion-exchange column Source 15Q (GE Healthcare) and eluted using a 140 ml linear gradient of 0–0.5 M NaCl. Finally, the protein TTD was purified using Superdex 200 Increase column (GE Healthcare) equilibrated in 10 mM Tris-HCl, pH 8.0, 200 mM NaCl, and 3 mM DTT.

**Site-directed mutagenesis.** Based on the structure of the TIRR-Tudor domain complex and sequence conservation analysis, five TIRR mutants (K10E, P105A, H106A, R107A, and L60Y/V143Y/F160A) and four tudor domain mutants (W1495A, Y1500A, D1521A, and Y1523A) were generated using QuikChange Site-Directed Mutagenesis Kit (Stratagene). The corresponding primers are listed in the Supplementary Tables 1 and 2. The sequence of constructs was confirmed by sequencing. The mutant proteins were purified following the protocol of the wild-type proteins.

**Crystallization and data collection.** To screen for crystallization of the TIRR and TTD complex, the two proteins were mixed at 1:1 ratio (TIRR: 10.9 mg/mL, TTD: 8.8 mg/mL). Preliminary crystallization conditions were obtained using the hanging drop vapor-diffusion method at 293K by mixing equal volumes of protein complex with reservoir solution containing 0.7 M ammonium tartrate and 0.1 M sodium acetate, pH 4.6. After optimization, the crystals were immersed in the cryoprotectant buffer consisting of the reservoir solution supplemented with 20% (v/v) glycerol and then flash-cooled in liquid nitrogen. The X-ray diffraction data was collected on the beamline BL17u1 at Shanghai Synchrotron Radiation facility (SSRF)[25]. The crystal belongs to space group C121 with the unit cell dimensions $a = 111.144$ Å, $b = 103.922$ Å, $c = 61.470$ Å, and $\beta = 95.46°$. The data was processed using the HKL-3000 software suite[26].

**Structure determination and refinement.** The crystal structure of the complex TIRR with TTD was solved through molecular replacement using Phaser[27]. The native TIRR from *Homo sapiens* was used as the search model (PDB ID: 3KVH). The structure building was carried out via the ARP/wARP software[28]. Manual model building and further refinement were performed repeatedly using COOT[29] and PHENIX[30]. The detailed statistics of data collection and structure refinement are summarized in Table 1. All the molecular graphics figures are prepared using PyMol (http://www.pymol.org).

**Table 1 Data collection and refinement statistics of the TIRR-TTD complex**

| | TDD–TIRR complex |
|---|---|
| **Data collection** | |
| Space group | C2 |
| Cell dimensions | |
| $a, b, c$ (Å) | 111.144 103.922 61.47 |
| $\alpha, \beta, \gamma$ (°) | 90.000 95.458 90.000 |
| Wavelength (Å) | 0.9789 |
| Resolution (Å) | 19.92–2.004 (2.076 - 2.004)[a] |
| $R_{merge}$(%) | 8.397(61.36) |
| $\langle I/\sigma(I)\rangle$ | 13.94 (3.97) |
| Completeness (%) | 97.35 (94.61) |
| Redundancy | 6.7 (6.8) |
| **Refinement** | |
| Resolution (Å) | 19.92–2.004 |
| No. reflections | 308418 |
| $R_{work}/R_{free}$ (%) | 18.99/21.89 |
| No. atoms | |
| Protein | 4130 |
| Ca | 2 |
| Water | 393 |
| B-factors | |
| Protein | 23.11 |
| Ligand/ion | 8.57 |
| Water | 31.99 |
| R.m.s deviations | |
| Bond lengths (Å) | 0.003 |
| Bond angles (°) | 0.93 |

[a]Statistics for the highest-resolution shell are shown in parentheses

**Sedimentation-velocity analytical ultracentrifugation.** Sedimentation-velocity measurements were carried out on an XL-I analytical ultracentrifuge (Beckman Coulter, Fullerton, CA) using a four-cell An-60 Ti rotor. TIRR and TTD were diluted to 53 μM and 35 μM, respectively, using the buffer containing 10 mM Tris-HCl, pH 8.0, 200 mM NaCl, and 3 mM DTT. The 20 μM TIRR was mixed with 20 μM TTD to analyze the aggregation state of the TIRR–TTD complex. The corresponding buffer was used as the reference solution. All samples were centrifuged at 60,000 rpm at 20 °C. Data collection was performed at 280 nm at 30 s intervals.

**Isothermal titration calorimetry.** Isothermal titration calorimetry (ITC) was carried out to measure the binding affinities between 53BP1 TTD and TIRR at 25 °C using MicroCal PEAQ-ITC. All protein samples were in the PBS buffer, pH 7.0 containing 140 mM NaCl. One-hundred and fifty micromolar 53BP1 TTD was injected into the calorimetric cell containing 15 μM wild-type TIRR or its mutants. As a control, the PBS buffer was titrated with 53BP1 TTD. In all, 8–10 μM TIRR was titrated with 80–100 μM 53BP1 TTD mutant to measure their binding affinities. To determine the binding affinity between H4K20me2 peptide (residues 12–25) and 53BP1 TTD, 750 μM H4K20me2 peptide was injected into the calorimetric cell containing 37.5 μM 53BP1 TTD. All titration consisted of a 0.4 μl pre-injection and consecutive 19 × 2 μl injections at 150 s intervals. The data obtained was processed with Origin software.

**Pull-down assays.** Two microgram GST-53BP1 TTD (N1459-C1634) was incubated with 2 μg His-TIRR and Glutathione Sepharose 4B beads (GE Healthcare) at 4 °C for 1 h with rotation. After washing with NETN-100 buffer (0.5% Nonidet P-40, 2 mM EDTA, 20 mM Tris-HCl, pH 8.0, 100 mM NaCl) four times, the samples were boiled in the SDS sample buffer. The elutes were analyzed by western blot with indicating antibodies. For the competition assays, biotin-H4K20me2 peptide (5 nmol) was incubated with GST-53BP1 TTD (5 nmol) and streptavidin beads in presence of 0, 0.5, 5, or 50 nmol TIRR-his, respectively, for 2 h at 4 °C. After washing with NETN-100 buffer, the elutes were analyzed by western blot. All antibodies used in this study are shown in Supplementary Table 3.

**Co-immunoprecipitation.** HEK-293T and U2OS cells were maintained in DEME medium with 10% fetal serum and cultivated at 37 °C in 5% CO2 (v/v). UWB1 (BRCA1-deficient) cells were maintained in MEGM medium with 6% fetal serum and cultivated at 37 °C in 5% CO2 (v/v). After transfection, cells were lysed with NETN-300 buffer (0.5% Nonidet P-40, 2 mM EDTA, 20 mM Tris-HCl, pH 8.0, 300 mM NaCl). Clear cell lysates were incubated with streptavidin sepharose beads (Thermo Fisher Scientific) for 2 h at 4 °C. After washing with NETN-100 buffer (0.5% Nonidet P-40, 2 mM EDTA, 20 mM Tris-HCl, pH 8.0, 100 mM NaCl) four times, the samples were boiled in the SDS sample buffer. The elutes were analyzed by western blot. All cell lines were purchased from American Type Culture Collection (ATCC).

**IR treatment and immunofluorescence staining.** U2OS cells or UWB1(BRCA1-deficient) were grown on glass coverslips, transfected, and irradiated with a 137Cs source at a dose of 10 Gy. After recovery for 4 h, cells were fixed in 3% paraformaldehyde for 15 min and permeabilized with 0.5% Triton X-100 in phosphate-buffered saline (PBS) for 10 min at room temperature. Samples were blocked with 8% goat serum and then incubated with the primary antibody for 1 h. Samples were washed for three times and incubated with the secondary antibody for 30 min. After PBS wash, the nuclei were stained by DAPI. The coverslips were mounted onto glass slides and visualized with OLYMPUS IX71 inverted fluorescence microscope. All the images were acquired with cellSens standard (Version 1.3) software under OLYMPUS IX71 inverted fluorescence microscope equipped with an UPlanSApo 60_/1.35 oil immersion objective at room temperature. Contrast and brightness settings were identically performed on all images in a given experiment.

**Cell viability assay.** Cells were seeded at 4000 cells/well into 96-well plates. Olaparib was serially diluted in medium and added into the wells at various concentrations. Six days later, 10 μl thiazolyl blue tetrazolium bromide solution (Sigma, 5 mg/ml, dissolved in PBS) was added to each well, and plates were incubated at 37 °C for 4 h. Then the supernatant was discarded and 50 μl DMSO was added to dissolve the formazan crystals before scanning with a luminescence microplate reader.

**Data availability.** The atomic coordinates and structure factors of the TIRR–53BP1 TTD complex have been deposited in the Protein Data Bank with the accession code 5ZCJ. Other data are available from the corresponding authors upon reasonable request.

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

## Acknowledgements

We thank the Beamline BL17u1 staff at the Shanghai Synchrotron Radiation facility for X-ray data collection experiments. This work was supported by the National Natural Science Foundation of China (Grant No. 81672794), the "100 Talents Plan" of Hebei Province (Grant No. E2016100014), Foundation of Hebei Educational committee (Grant No. YQ2014007 and SLRC2017023) and the National Institute of Health (CA130899, CA132755, CA187209, GM108647). X.Y is recipient of the Scholar Award from Leukemia and Lymphoma Society (1315–25).

## Author contributions

X.Y. designed the project. X.L. administrated the project. J.W., Y.C., R.X., G.Y., M.K., M.W., Y.M., and C.W. performed the experiments. Z.Y., X.Y., and X.L. analyzed the data. X.Y. and X.L. wrote the manuscript.

## Additional information

**Competing interests:** The authors declare no competing interests.

