## [Peer Review File · Nature Communications]

Reviewers' comments:

Reviewer #1 (Remarks to the Author):

In this manuscript Wang et al. have studied the structure and function of the TIRR-53BP1 complex. 53BP1 is a well known DNA damage signaling protein that interacts with post-translational modifications on chromatin surrounding sites of DNA double strand breaks to facilitate repair by the NHEJ pathway. TIRR is a recently discovered inhibitor of 53BP1. The authors here determine the crystal structure of the 53BP1 tandem tudor domain bound to TIRR. The structure not only shows the mechanism of interaction but suggests that this interaction will block 53BP1 interactions with H4K20me2, a major mark associated with the recruitment of 53BP1 to damaged chromatin. They go on to use mutagenesis to show the importance of the binding interface for TIRR-53BP1 interactions in vitro and in cells, and show that mutations that disrupt the interaction also interfere with 53BP1 recruitment to damaged DNA and downstream DNA damage signaling in human cells. Overall this is a solid paper, reporting on an interesting, newly identified regulatory complex in double strand break signaling which will be of considerable to those in this field. My only major concern is the writing – significant editing will be required to improve the English for clarity and readability.

Comments:

- I/sI in the highest resolution bin is ~4. Could the authors have extended the resolution higher?
- Panel 3F – probably not needed since none of the mutants were found to bind. The authors could instead say that these results are consistent with the pull downs and refer to the supplemental figure S3.
- Pg. 4 and 9 – the authors state that NUDT16 is an enzyme that can hydrolyse a phosphodiester bond. A little more background information on NUDT16 should be given in the introduction.

Reviewer #2 (Remarks to the Author):

In the manuscript entitled “Structure basis for the inhibition of the methyl-lysine binding function of 53BP1 by TIRR” by Wang and Colleagues, the authors successfully obtain a co-crystal structure composing of TIRR and 53BP1-TTD domain, and map and verify the critical residues for their interactions by using different approaches including ITC and co-IP. The introduction of mutations in these residues, either on 53BP2-TTD or TIRR, abolish their interaction and compromises 53BP1's cellular functions. Although the crystallography result is new, some of the crucial sites are already known for the interaction of two proteins and most of the interaction functions of TIRR on 53BP1's damage responses have been reported previously. I do not find the current study have significant novelty and provide further advances on our understanding of 53BP1 regulation. In my opinion, the study is a bit too pre-mature for the publication of the current journal. Further important questions need to be addressed, such as what is the functions of TIRR homo-dimerization and of TIRR/NUDT16 hetero-dimerization on 53BP1 stability and DNA damage responses. Besides, a more crucial question is how TIRR/53BP1 dissociation is triggered by the recruitment of RIF1.

Figure 4B, 4C, 5A and 5B should also show counting on the parental cells without any over-expression of TIRR as a control. Statistics analyses are missing too.

Figure 5C, log scales should be used in survival curves.

In page 10, a typing mistake of “BRCA1-defient cells”

In general, “ectopic expression” should be replaced by “ectopic over-expression” as the experiments were done by transient transfection and the interference of 53BP1 functions seem

dependent on the over-production of TIRR protein.

Response to reviewers' comments:

We are very grateful to the constructive suggestions from both reviewers. Following the reviewers' suggestions, we have performed additional experiments and modified our manuscript. As listed below, we have point-by-point addressed all the concerns raised from both reviewers.

Reviewer #1

General comments:

“In this manuscript Wang et al. have studied the structure and function of the TIRR-53BP1 complex...My only major concern is the writing – significant editing will be required to improve the English for clarity and readability.”

Thank you for the positive comments. The manuscript has been edited by other researchers in the field. We have tried our best to correct the typos and errors.

Specific Comments:

1. *“I/sI in the highest resolution bin is ~4. Could the authors have extended the resolution higher?”*

Answer: Thank you for the suggestion. During the data collection, the diffraction has already been extended to the detector edge. Because the data quality for the highest resolution bin is good enough (completeness is 94.61% and Rmerge is 61.36%), we did not perform the resolution cut-off. Thus, the I/sI value for the highest resolution bin is 3.97.

2. *“Panel 3F – probably not needed since none of the mutants were found to bind. The authors could instead say that these results are consistent with the pull downs and refer to the supplemental figure S3.”*

Answer: We agree with the reviewer and have deleted Fig. 3F. The original ITC results have been included in the supplemental Figure S3.

3. *“Pg. 4 and 9 – the authors state that NUDT16 is an enzyme that can hydrolyse a phosphodiester bond. A little more background information on NUDT16 should be given in the introduction.”*

Answer: Thank you for the suggestion! We included a brief introduction of NUDT16 with focusing on its enzymatic activity. It will allow readers to understand the possible biological function of NUDT16 in the context of DNA damage repair (Page4 line8-9; and Page9 line6-7). However, these enzymatic activities have not been further confirmed by other groups. Thus, additional studies are needed to reveal the molecular mechanism and biological function of NUDR16 in future.

Reviewer #2

General comments:

“In the manuscript entitled “Structure basis for the inhibition of the methyl-lysine binding function of 53BP1 by TIRR” by Wang and Colleagues, the authors successfully obtain a co-crystal structure composing of TIRR and 53BP1-TTD domain, and map and verify the critical residues for their interactions by using different approaches including ITC and co-IP. The introduction of mutations in these residues, either on 53BP2-TTD or TIRR, abolish their interaction and compromises 53BP1’s cellular functions. Although the crystallography result is new, some of the crucial sites are already known for the interaction of two proteins and most of the interaction functions of TIRR on 53BP1’s damage responses have been reported previously. I do not find the current study have significant novelty and provide further advances on our understanding of 53BP1 regulation. In my opinion, the study is a bit too pre-mature for the publication of the current journal. Further important questions need to be addressed, such as what is the functions of TIRR homo-dimerization and of TIRR/NUDT16 hetero-dimerization on 53BP1 stability and DNA damage responses. Besides, a more crucial question is how TIRR/53BP1 dissociation is triggered by the recruitment of RIF1.”

Answer: Thank you for the constructive suggestions. We agree with the reviewer that the interaction between TIRR and 53BP1 has been reported^{1,2}. However, the structure of the complex has not been solved yet. Nor have the details of the interaction been characterized. Here, we show the first evidence of the structure of the complex, which reveals novel and detailed binding sites on both TIRR and the tudor domain of 53BP1. Moreover, it has been shown that TIRR is overexpressed in human tumors and regulates tumor cells sensitivity to PARP inhibitor treatment¹. Thus, our structure analysis provides the invaluable resources and opportunities for other researchers to develop chemical probes for clinical cancer treatment in future. Thus, the structure analysis of this complex will generate impact in the field.

Moreover, as this reviewer mentioned, we have revealed the dimer formation of TIRR. As suggested by the reviewer and based on the structural analyses, we generated different mutations in TIRR to abolish the TIRR homodimer and TIRR/NUDT16 heterodimer. With the pull down screening, we found that triple-mutation (L60Y/V143Y/F160A) at the dimer interface abolished the dimer formation (Supplemental Figure 9A, E and F). Because of the broad interface between the dimer, the single mutations could not disrupt the dimer formation. Moreover, these mutations did not affect the interaction with 53BP1 because the dimer interface is far away from the interaction sites with 53BP1 (Supplemental Figure 9B and 9E). However, disrupting the dimer formation destabilized 53BP1 in the cell, suggesting that the dimer formation plays a key role to maintain the complex stability³ (Supplemental Figure 9C). Thus, the results further strengthen our conclusions in the manuscript. And we have included the analysis of the TIRR dimer in the result section (Page 11-13).

In addition, the possible mechanism of dissociation of the TIRR/53BP1 complex could be very complicated. Because RIF1 is a relatively big size protein (2472 amino acid residues), the structure of RIF1 has not been solved. It is also unclear if RIF1 directly affects the dissociation of the TIRR/53BP1 complex. RIF1 interacts with the N-terminal S/TQ domain of 53BP1⁴⁻⁶. Since N-terminal S/TQ of 53BP1 is phosphorylated by PI3-like kinases in response to DNA damage⁴⁻⁷, it is possible that these phosphorylation events induce the overall conformation

change in 53BP1, which causes the release of TIRR. However, this is only one possibility that causes the dissociation of the complex. There are also other possibilities. For example, we noticed that 53BP1 was sumoylated in response to DNA damage. Interestingly, with unbiased proteomic analysis, one key sumoylation site has been mapped at Lys1563⁸, which is in the Tudor domain. Although K1563 is not at the interaction sites with TIRR, the bulky sumoylation at K1563 may abolish the interaction with TIRR. However, unlike TIRR, H4K20me is only recognized by the small pocket with minimal contact in the Tudor domain. The sumoylation at K1563 may not affect the interaction with H4K20me. Because the analysis of the mechanism by which induces the dissociation of the 53BP1-TIRR complex is well beyond the current research scope on how the 53BP1-TIRR complex abolishes the interaction with H4K20me, we included the results only for the reviewer. We plan to further characterize this interesting phenomenon and examine the detailed molecular mechanism of the dissociation of the 53BP1-TIRR complex by sumoylation in the tudor domain or phosphorylation at the N-terminus of 53BP1 as a separate independent research project.

Specific Comments:

1. *“Figure 4B, 4C, 5A and 5B should also show counting on the parental cells without any over-expression of TIRR as a control. Statistics analyses are missing too.”*

Answer: As suggested by the reviewer, we have included the parental cells as the control in Figure 4B, 4C, 5A and 5B. Moreover, statistical analyses were included in the revised manuscript (Figure 4 and 5).

2. *“Figure 5C, log scales should be used in survival curves.”*

Answer: In the original data, we used MTT assays to examine the cell viability in a relatively short time frame. Here, we re-examined the role of TIRR in DNA damage repair with clonogenic assays. The data were shown in the log scales in the revised Figure 5C.

3. *“In page 10, a typing mistake of “BRCA1-defient cells””*

Answer: Thank you! We have corrected the typo.

4. *“In general, “ectopic expression” should be replaced by “ectopic over-expression” as the experiments were done by transient transfection and the interference of 53BP1 functions seem dependent on the over-production of TIRR protein.”*

Answer: As suggested, we have corrected to “ectopic over-expression” in the text.

Reference:

1. Drane, P. et al. TIRR regulates 53BP1 by masking its histone methyl-lysine binding function. *Nature* **543**, 211-216 (2017).
2. Zhang, A., Peng, B., Huang, P., Chen, J. & Gong, Z. The p53-binding protein 1-Tudor-interacting repair regulator complex participates in the DNA damage response. *J Biol Chem* **292**, 6461-6467 (2017).
3. Drane, P. & Chowdhury, D. TIRR and 53BP1- partners in arms. *Cell Cycle* **16**, 1235-1236 (2017).
4. Chapman, J.R. et al. RIF1 is essential for 53BP1-dependent nonhomologous end joining and suppression of DNA double-strand break resection. *Mol Cell* **49**, 858-71 (2013).
5. Escribano-Diaz, C. et al. A cell cycle-dependent regulatory circuit composed of 53BP1-RIF1 and BRCA1-CtIP controls DNA repair pathway choice. *Mol Cell* **49**, 872-83 (2013).
6. Silverman, J., Takai, H., Buonomo, S.B., Eisenhaber, F. & de Lange, T. Human Rif1, ortholog of a yeast telomeric protein, is regulated by ATM and 53BP1 and functions in the S-phase checkpoint. *Genes Dev* **18**, 2108-19 (2004).
7. Callen, E. et al. 53BP1 mediates productive and mutagenic DNA repair through distinct phosphoprotein interactions. *Cell* **153**, 1266-80 (2013).
8. Impens, F., Radoshevich, L., Cossart, P. & Ribet, D. Mapping of SUMO sites and analysis of SUMOylation changes induced by external stimuli. *Proc Natl Acad Sci U S A* **111**, 12432-7 (2014).